

# Addition of L-cysteine to the N- or C-terminus of the all-D-enantiomer [D(KLAKLAK)2] increases antimicrobial activities against multidrug-resistant *Pseudomonas aeruginosa, Acinetobacter baumannii* and *Escherichia coli*

Maki K. Ohno[1,*], Teruo Kirikae[2], Eisaku Yoshihara[3], Fumiko Kirikae[2] and Isao Ishida[1,*]

[1] Faculty of Pharmaceutical Sciences, Teikyo Heisei University, Tokyo, Japan
[2] Department of Microbiology, Juntendo University School of Medicine, Tokyo, Japan
[3] Tokai University School of Medicine, Kanagawa, Japan
[*] These authors contributed equally to this work.

## ABSTRACT

**Background**. Antimicrobial peptides have a broad spectrum of antimicrobial activities and are attracting attention as promising next-generation antibiotics against multidrug-resistant (MDR) bacteria. The all-d-enantiomer [D(KLAKLAK)2] has been reported to have antimicrobial activity against *Escherichia coli* and *Pseudomonas aeruginosa*, and to be resistant to protein degradation in bacteria because it is composed of D-enantiomer compounds. In this study, we demonstrated that modification of [D(KLAKLAK)2] by the addition of an L-cysteine residue to its N- or C- terminus markedly enhanced its antimicrobial activities against Gram-negative bacteria such as MDR *Acinetobacter baumannii, E. coli*, and *P. aeruginosa*.

**Methods**. The peptides [D(KLAKLAK)2] (DP), DP to which L-cysteine was added at the N-terminus C-DP, and DP to which L-cysteine was added at the C-terminus DP-C, were synthesized at >95% purity. The minimum inhibitory concentrations of peptides and antibiotics were determined by the broth microdilution method. The synergistic effects of the peptides and the antibiotics against MDR *P. aeruginosa* were evaluated using the checkerboard dilution method. In order to assess how these peptides affect the survival of human cells, cell viability was determined using a Cell Counting Kit-8.

**Results**. C-DP and DP-C enhanced the antimicrobial activities of the peptide against MDR Gram-negative bacteria, including *A. baumannii, E. coli,* and *P. aeruginosa*. The antimicrobial activity of DP-C was greater than that of C-DP, with these peptides also having antimicrobial activity against drug-susceptible *P. aeruginosa* and drug-resistant *P. aeruginosa* overexpressing the efflux pump components. C-DP and DP-C also showed antimicrobial activity against colistin-resistant *E. coli* harboring *mcr-1*, which encodes a lipid A modifying enzyme. DP-C showed synergistic antimicrobial activity against MDR *P. aeruginosa* when combined with colistin. The $LD_{50}$ of DP-C against a human cell line HepG2 was six times higher than the MIC of DP-C against MDR *P. aeruginosa*. The $LD_{50}$ of DP-C was not altered by incubation with low-dose colistin.

Corresponding author
Maki K. Ohno, m.ohno@thu.ac.jp

**Conclusion**. Attachment of an L-cysteine residue to the N- or C-terminus of [D(KLAKLAK)$_2$] enhanced its antimicrobial activity against *A. baumannii*, *E. coli*, and *P. aeruginosa*. The combination of C-DP or DP-C and colistin had synergistic effects against MDR *P. aeruginosa*. In addition, DP-C and C-DP showed much stronger antimicrobial activity against MDR *A. baumannii* and *E. coli* than against *P. aeruginosa*.

## INTRODUCTION

The emergence and spread of multidrug-resistant (MDR) Gram-negative pathogens has become a serious public health problem worldwide. A global priority list of antibiotic-resistant bacteria published by the World Health Organization (WHO) to guide research, discovery, and development of new antibiotics listed carbapenem-resistant *Acinetobacter baumannii*, *Pseudomonas aeruginosa* and carbapenem-resistant and third-generation cephalosporin-resistant *Enterobacteriaceae* as first priority pathogens (*Shai, 2002*).

Antimicrobial peptides (AMPs) are produced by various host organisms and partially contribute to the host's innate immunity (*Plesniak et al., 2004*; *Onuchic, Jennings & Ben-Jacob, 2013*). These peptides exhibit potent antimicrobial activities against a wide range of microorganisms, including viruses, bacteria, protozoa, and fungi (*Shai, 2002*). AMPs are chemically amphiphilic polycationic peptides, generally comprising 6–50 amino acid residues, and they constitute a unique and diverse group of molecules (*Peters, Shirtliff & Jabra-Rizk, 2010*). AMPs are classified according to their secondary structures including mixtures of $\alpha$-helices, $\beta$-sheets, loops, and extended peptides. Most of these peptides are thought to bind to the cytoplasmic membrane, forming micelle-like aggregates that destroy the membrane (*Peters, Shirtliff & Jabra-Rizk, 2010*). These peptides orient parallel to the interface, and associate with the membrane surface. After reaching a threshold concentration on the bilayer surface, they aggregate promoting channel formation through the bilayer and disrupt the membrane (*Matsuzaki et al., 1995*; *Plesniak et al., 2004*; *Onuchic, Jennings & Ben-Jacob, 2013*). Because of their mechanism of action, AMPs show antimicrobial activities against MDR as well as drug-susceptible Gram-negative bacteria. Thus, AMPs are attracting attention as promising next-generation antibiotics for the treatment of MDR bacterial infections (*Hancock & Lehrer, 1998*; *Marr, Gooderham & Hancock, 2006*).

The all-D-enantiomer, [D(KLAKLAK)$_2$] is an amphipathic lysine-leucine-rich $\alpha$-helical peptide with high antimicrobial activity against *Escherichia coli* and *P. aeruginosa*, but with low toxicity against mouse 3T3 cells (*Javadpour et al., 1996*; *McGrath et al., 2013*). [D(KLAKLAK)$_2$] orients parallel to the interface and associates with the outer membrane surface. After reaching a threshold concentration on the outer membrane surface, the peptides aggregate to promote channel formation through the bilayer (*Matsuzaki et al., 1995*; *Plesniak et al., 2004*; *Onuchic, Jennings & Ben-Jacob, 2013*). [D(KLAKLAK)$_2$] was reported to selectively interfere with the bilayer of the outer membranes of *E. coli* and
*P. aeruginosa*, leading to cell death by membrane disruption and loss of membrane potential. [$_D$(KLAKLAK)$_2$] has shown antimicrobial activity against Gram-negative bacteria but not Gram-positive bacteria, as this peptide was unlikely to disrupt the thick peptidoglycan layer of the latter (*McGrath et al., 2013*). One of the strategies used to protect AMPs from protease degradation was sequence modification of D-amino acids to replace L-amino acids (*Choi et al., 1993*; *Braunstein, Papo & Shai, 2004*; *Lee & Lee, 2008*). Because this peptide is an all-D-enantiomer, it is highly resistant to proteolysis in bacteria and has low immunogenicity. This stability and low immunogenicity may prolong its half-life and enhance its efficacy at low doses in vivo. Initially we added L-cysteine to the N- or C-terminus of [$_D$(KLAKLAK)$_2$] in order to conjugate with protein such as a single chain antibody. The side chain of cysteine contains sulfhydryl group, which can make a covalent coupling with an amino group of the protein via a cross-linker molecule such as sulfo-SMCC (sulfosuccinimidyl 4-(N-maleimidomethyl) cyclohexane-1-carboxylate). The cysteine-rich AMPs were isolated from leguminous plants and the granular hemocytes of mangrove crabs (*Sivakamavalli, Nirosha & Vaseeharan, 2015*; *Maróti, Downie & Kondorosi, 2015*). The functionalized textiles and nasal prongs modified with L-cysteine exhibited antimicrobial activity against *Staphylococcus aureus* and *Klebsiella pneumoniae* (*Gouveia, Sá & Henriques, 2012*; *Caldeira et al., 2013*; *Xu et al., 2017*; *Odeberg et al., 2018*).

Modification of [$_D$(KLAKLAK)$_2$] by the addition of an L-cysteine residue to its N- or C- terminus markedly enhanced its antimicrobial activities against Gram-negative bacteria such as *P. aeruginosa*, *E. coli*, and *A. baumannii*. The present study describes the antimicrobial activities of the modified peptides against clinical isolates of MDR *P. aeruginosa*, *A. baumannii*, and *E. coli*, and its synergistic effects with low dose colistin.

## MATERIALS & METHODS

### Peptide design and synthesis

Peptides [(D-Lys-D-Leu-D-Ala-D-Lys-D-Leu-D-Ala-D-Lys)$_2$] (DP) (*McGrath et al., 2013*), DP to which L-cysteine was added at the N-terminus, [L-Cys-(D-Lys-D-Leu-D-Ala-D-Lys-D-Leu-D-Ala-D-Lys)$_2$] (C-DP), and DP to which L-cysteine was added at the C-terminus [(D-Lys-D-Leu-D-Ala-D-Lys-D-Leu-D-Ala-D-Lys)$_2$-L-Cys] (DP-C), were synthesized with >95% purity (Scrum Inc., Tokyo, Japan). DP-C was dimerized by heating at 60 °C for 30 min (DP-C dimer) to convert cysteine to cystine. The purity of the synthetic peptides was checked on a SunFire C18 column (100 Å, 5 μm, 4.6 mm inner diameter × 250 mm; Waters, Milford, MA, USA). Each peptides were gradiently eluted with solution A (water containing 0.1% trifluoroacetic acid) and solution B (acetonitrile containing 0.1% trifluoroacetic acid) at a flow rate of 1.0 mL/min. The elution program for DP and C-DP was as follows: at 0 min, 10% of B; at 20 min, 60% of B. The elution program for DP-C was as follows: at 0 min, 0% of B; at 20 min, 100% of B. The separated components were detected at 220 nm. The DP-C dimer was separated on a COSMOSIL 5C18-AR-300 reversed phase column (4.6 mm inner diameter × 250 mm; Nacalai Tesque, Kyoto, Japan), using an automated HPLC system (LC-2010AHT; Shimadzu, Kyoto, Japan). The reaction products were gradiently eluted with solution A (water containing 0.086% trifluoroacetic

acid) and solution B (acetonitrile containing 0.086% trifluoroacetic acid) at a flow rate of 1.0 mL/min. The elution program for DP-C dimer was as follows: at 0 min, 20% of B; at 20 min, 50% of B. The peptide masses were determined by MALDI-TOF MS on a microflex (Bruker, Billerica, MA, USA).

## Bacterial strains

MDR *P. aeruginosa* NCGM2.S1 (*Miyoshi-Akiyama et al., 2011*) and drug-susceptible *P. aeruginosa* PAO1 (*CLSI, 2018*), *P. aeruginosa* OCR1 (*Poole et al., 1996*), and *P. aeruginosa* PAO4290 (*Yoneyama et al., 1997*) were grown in Luria Bertani (LB) broth (BD Japan, Tokyo, Japan) or on LB plates containing 15 g/L agar at 37 °C. Drug-susceptible *E. coli* ATCC 25922, a clinical isolate of *E.coli* NCCHD1261-5 (*Uchida et al., 2018*), and *S. aureus* ATCC 25923 were grown at 37 °C in tryptic soy broth (BD). Drug-susceptible *A. baumannii* ATCC 15308, a clinical isolate of MDR *A. baumannii* IOMTU433 (*Tada et al., 2015*) (GenBank accession no. AP014649), a clinical isolate of MDR *A. baumannii* NCGM237 (*Tada et al., 2015*) (GenBank accession no. AP013357), a clinical isolate of MDR *A. baumannii* NCGM253 (*Tada et al., 2015*) (GenBank accession no. AB823544), *K. pneumoniae* ATCC-BAA-2146, *K. pneumoniae* ATCC15380 (*Reading & Cole, 1977*), and *Serratia marcescens* NBRC102204$^T$ were grown at 37 °C in Difco$^{TM}$ Nutrient broth (BD).

## Drug susceptibility testing

The minimum inhibitory concentrations (MICs) of peptides and antibiotics, including meropenem, amikacin, ofloxacin, and colistin, were determined by the broth microdilution method according to Clinical Laboratory Standards Institute (CLSI) guidelines (*CLSI, 2018*). Bacterial strains were inoculated at $5 \times 10^5$ CFU/mL per well into 96-well round-bottom microtiter plates (Watson Bio Lab, Kobe, Japan) containing an equal volume of serially diluted peptides or antibiotics. Three independent experiments were performed to confirm reproducibility.

The synergistic effects of the peptides and the antibiotics amikacin, colistin, meropenem, ofloxacin, and rifampicin against *P. aeruginosa* NCGM2.S1 were evaluated using the checkerboard dilution method. The peptides were two-fold serially diluted to final concentrations ranging from 0.125- to 2-times the MIC longitudinally in 96-well round-bottom microtiter plates. Subsequently, antibiotic was two-fold serially diluted to final concentrations ranging from 0.125- to 2-times the MIC transversely into the plates. NCGM2.S1 was inoculated at $5 \times 10^5$ CFU/mL per well at a volume equal to that of the diluted peptide and antibiotic. Three independent experiments were performed to confirm reproducibility. The synergistic effect of the peptides and antibiotics was assessed by determining the fractional inhibitory concentration (FIC) index (*Berenbaum, 1978*), using the formula:

$$FIC = \frac{\text{MIC of peptide} \in \text{combination}}{\text{MIC of peptide alone}}$$
$$+ \frac{\text{MIC of antimicrobial agent} \in \text{combination}}{\text{MIC of antimicrobial agent alone}}.$$

An FIC index ≤0.5 was defined as synergistic, an FIC index >0.5 to 4.0 was defined as additive or unrelated, and an FIC index >4.0 was defined as antagonistic.

### Cytotoxicity tests

The human hepatoblastoma cell line, HepG2 (ATCC HB-8065), was obtained from American Type Culture Collection and cultured in Dulbecco's modified Eagle's medium supplemented with 10% fetal bovine serum (FBS). HepG2 cells were seeded at 3,000 cells/well in 96-well cell culture-treated flat bottom microtiter plates (Falcon, Corning, NY, USA). The cells were incubated at 37 °C for 48 h in an atmosphere containing 5% $CO_2$, followed by the addition of peptides at a final concentration of 0–256 µg/mL, or colistin, at a final concentration of 0–3,000 µg/mL, by serial dilution. The plates were incubated with 0.2% FBS in 5% $CO_2$ at 37 °C for 48 h; under these conditions, HepG2 cells were alive, but they did not grow. Cell viability was determined using a Cell Counting Kit-8 (Dojin, Tokyo, Japan), and colorimetric changes were determined at $OD_{450}$ with a microplate reader (Corona Electric Co. Ltd., Ibaraki, Japan). $LD_{50}$ was defined as the concentration of peptides or colistin that resulted in 50% cell viability. Three independent experiments were performed to confirm reproducibility.

### Statistical analysis

The Mann–Whitney U test was used to compare the MIC values of C-DP, DP-C, and DP-C dimer with DP in *P. aeruginosa* and *A. baumannii*. *P*-values less than 0.05 were considered statistically significant. Cell survival was expressed as a percentage of the control were obtained as mean ± standard deviation of three independent experiments done in three replicates for each treatment. Significant differences of cell survival rate between each concentration and the control were statistically evaluated by Student's *t*-test.

## RESULTS

### Addition of L-cysteine to the N- or C- terminus enhanced the antimicrobial activity of the original peptide

HPLC analysis indicated that the purities of synthetic DP, C-DP, and DP-C were 100, 95.52, and 98.68%, respectively (Figs. S1–S3). Additionally, the masses of DP, C-DP, and DP-C by MALDI-TOF MS analysis, were 1525.444, 1626.595, and 1627.151, respectively that matched well with the theoretical molecular weights (1524.0, 1627.1, and 1627.1) (Figs. S1–S3). Similarly, the formation of DP-C dimer was assessed by HPLC and MALDI-TOF MS. The HPLC analysis of heat treated DP-C showed one large peak estimated as DP-C dimer and one small peak estimated as DP-C monomer the peak area ratio was 6.3:1 (Fig. S4). The MALDI-TOF MS result of heat treated DP-C was 3252.7, which was consistent with the estimated molecular weight of DP-C dimer (Fig. S4). The grand average hydropathy (GRAVY) values were −0.07 for DP and 0.1 each for C-DP and DP-C, indicating that the addition of L-cysteine affected the hydrophobicity of DP.

Assessment of antibiotic susceptibility showed that drug-susceptible *P. aeruginosa* PAO-1 (*CLSI, 2018*) and PAO4290 expressing a normal level of MexAB-OprM (*Yoneyama et al., 1997*), were susceptible to all antibiotics tested; MDR *P. aeruginosa* NCGM2.S1

**Table 1** **MICs of antibiotics and antimicrobial peptides against *Pseudomonas aeruginosa* strains.** The Mann-Whitney $U$ test was used to compare the MIC values of C-DP, DP-C, and DP-C dimers with DP in *P. aeruginosa*. $P$-values less than 0.05 were considered statistically significant (* $p < 0.05$).

| Strains of *P. aeruginosa*[a] | MIC (µg/mL) | | | | | | | |
|---|---|---|---|---|---|---|---|---|
| | Antibiotics | | | | Antimicrobial peptides | | | |
| | Amikacin | Colistin | Meropenem | Ofloxacin | DP | C-DP* | DP-C* | DP-C Dimer*[b] |
| PAO-1 | 4 | 1 | 2 | 0.5 | 300 | 64 | 16 | 16 |
| NCGM2.S1 | 128 | 1 | >512 | 64 | 300 | 128 | 32 | 32 |
| OCR1 | 8 | 1 | 8 | 2 | >300 | 128 | 16 | 16 |
| PAO4290 | 4 | 1 | 1 | 1 | 300 | 128 | 16 | 16 |

**Notes.**

[a] *P. aeruginosa* strains used in this study were wild type PAO-1 (*CLSI, 2018*), the MDR clinical strain NCGM2.S1 (*Miyoshi-Akiyama et al., 2011*), the OprM overexpressing mutant OCR1 (*Poole et al., 1996*) and PAO4290 (*Yoneyama et al., 1997*) which expressed a wild-type level of MexAB-OprM.

[b] Generated by heating DP-C at 60 °C for 30 min to convert cysteine to cystine.

(*Miyoshi-Akiyama et al., 2011*) was susceptible to colistin, but resistant to amikacin, meropenem and ofloxacin; and OCR1, a *nalB* MDR mutant that overproduces the outer membrane protein OprM (*Poole et al., 1996*) was susceptible to amikacin and colistin, intermediately susceptible to ofloxacin, but resistant to meropenem (Table 1). DP showed antimicrobial activity against PAO-1, with an MIC of 300 µg/mL, consistent with previous findings (*McGrath et al., 2013*). DP also had antimicrobial activity against NCGM2.S1 and PAO4290, with MICs of 300 µg/mL, but DP showed no antimicrobial activity against OCR1 within the tested concentration range. C-DP had greater antimicrobial activities than DP against all of these strains (The Mann–Whitney U test, $p < 0.05$), with MICs of 64–128 µg/mL, and DP-C had greater antimicrobial activities than C-DP and DP (The Mann–Whitney U test, $p < 0.05$), with MICs 16–32 µg/mL (Table 1). The DP-C dimer also had antimicrobial activities and showed MICs identical to DP-C against all the strains tested. These results indicate that the addition of L-cysteine to the N- or C-terminus of $[_D(KLAKLAK)_2]$ increased its antimicrobial activity.

DP showed antimicrobial activity against four *A. baumannii* strains, with MICs of 64 to 300 µg/mL, and against a clinical isolate of colistin- and carbapenem-resistant *E. coli* NCCHD1261-5 co-harboring *mcr-1* and *bla*$_{NDM-5}$ genes (*Uchida et al., 2018*), with an MIC of 64 µg/mL. In contrast, DP was inactive against drug-susceptible *E. coli* ATCC 25922, two *K. pneumoniae* strains, *S marcescens* NBRC102204 and *S. aureus* ATCC 25923 (Table 2). C-DP and DP-C showed higher antimicrobial activities than DP (The Mann–Whitney U test, $p < 0.05$), with MICs of 4–8 µg/mL against the four *A. baumannii* strains and MICs of 4 to 16 µg/mL against the two *E. coli* strains. C-DP and DP-C showed antimicrobial activity against carbapenem-resistant *K. pneumonia* e ATCC BAA-2146 harboring *bla*$_{NDM-1}$, with both having MICs of 16 µg/mL, but not against penicillin-resistant, $\beta$-lactamase-producing *K. pneumonia* e ATCC 15380, *S. marcescens* NBRC102204 and *S. aureus* ATCC 25923 (Table 2).

## Synergistic effects of peptides and antibiotics

The combinations of DP, C-DP, and DP-C with colistin had synergistic effects on antimicrobial activity (Table 3). For example, the growth of *P. aeruginosa* NCGM2.S1

**Table 2  MICs of antimicrobial peptides against strains of bacteria.** The Mann-Whitney $U$ test was used to compare the MIC values of C-DP and DP-C with DP in species. $P$-values less than 0.05 were considered statistically significant (* $p < 0.05$).

| Strains[a] | MIC (μg/mL) | | | Genes or mutations associated with drug resistance | | |
|---|---|---|---|---|---|---|
| | DP | C-DP | DP-C | β-lactamase(s) | 16S rRNA methylase | Colistin-resistance gene |
| *Acinetobacter baumannii**  | | | | | | |
| ATCC15308 | 300 | 8 | 8 | | | |
| IOMTU433 | 64 | 4 | 8 | $bla_{NDM-1}$, $bla_{OXA-23}$, $bla_{PER-7}$ | | |
| NCGM237 | 128 | 4 | 8 | $bla_{OXA-23}$ | armA | |
| NCGM253 | 128 | 4 | 8 | $bla_{OXA-72}$ | | |
| *Escherichia coli* | | | | | | |
| ATCC 25922 | >300 | 16 | 8 | | | |
| NCCHD1261-5 | 64 | 4 | 8 | $bla_{NDM-5}$ | | mcr-1 |
| *Klebsiella pneumoniae* | | | | | | |
| ATCC 15380 | >128 | >128 | 128 | | | |
| ATCC BAA-2146 | >128 | 16 | 16 | $bla_{NDM-1}$ | | |
| *Serratia marcescens* | | | | | | |
| NBRC102204 | >256 | >256 | >256 | | | |
| *Staphylococcus aureus* | | | | | | |
| ATCC 25923 | >128 | >128 | >128 | | | |

**Notes.**
[a] *A. baumannii* strains were wild-type strain ATCC 15308 and multi-drug resistant strains IOMTU433 (*Tada et al., 2015*) (GenBank accession no. AP014649), NCGM237 (*Tada et al., 2015*) (GenBank accession no. AP013357) and NCGM253 (*Tada et al., 2015*) (GenBank accession no. AB823544). E. coli strains were wild-type strain ATCC25922 and multi-drug resistant strain NCCHD1261-5 (*Uchida et al., 2018*). *K. pneumoniae* strains were multidrug-resistant strain ATCC15380 (*Reading & Cole, 1977*) and the penicillin resistant strain ATCC-BAA-2146, a resistance caused by the production of β-lactamase. The *S. marcescens* strain NBRC102204 and the *S. aureus* strain ATCC 25923 were wild-type strain.

was inhibited by a combination of one-sixteenth the MIC of DP (19 μg/mL), and one-fourth the MIC of colistin (0.25 μg/mL), by a combination of one-fourth the MIC of C-DP (32 μg/mL) and one-eighth the MIC of colistin (0.125 μg/mL), and by a combination of one-eighth the MIC of DP-C (4.0 μg/mL) and one-fourth the MIC of colistin (0.25 μg/mL). Although the AMPs that induced susceptibility to rifampicin were reported in clinical MDR isolates of *P. aeruginosa* (*Baker et al., 2019*), DP-C only slightly enhanced the susceptibility to rifampicin. The combination of C-DP with meropenem or ofloxacin had additive effect on antimicrobial activity. Synergistic effects were not observed when DP-C was combined with amikacin.

## Cytotoxicity of antimicrobial peptides to HepG2 cells

The $LD_{50}$ values of each peptide in HepG2 cells were >256, >256, 192, and 2100 μg/mL for DP, C-DP, DP-C, and colistin, respectively (Figs. S5 and S6). The raw data of Figs. S5 and S6 are available in Files S2 and S3, respectively. Since the antimicrobial activity of these peptides against *P. aeruginosa* were synergistic with colistin, we examined whether a combination of these peptides and colistin was more toxic than the peptide alone. The combination of DP-C and colistin dose-dependently induced the death of HepG2 cells (Fig. 1), with more than 50% of the cells dying at 256 μg/ml DP-C and 25.6 μg/mL colistin. The cytotoxicity of the peptide was not enhanced by combination of low doses of colistin. There was no cytotoxicity to HepG2 at 4 μg/mL DP-C and 0.4 μg/mL colistin, which

**Table 3** **FIC index of combinations of antibiotics and antimicrobial peptide against *P. aeruginosa*.** The synergistic effects of DP, C-DP, or DP-C and antibiotics against *P. aeruginosa* NCGM2.S1 were analyzed by the checkerboard dilution method and the FIC index for each combination was calculated.

| Combination | FIC index[a] | Interpretation |
| --- | --- | --- |
| DP-C + Amikacin | 1.5 | Additive/Indifference |
| DP + Colistin | **0.31** | **Synergy** |
| C-DP + Colistin | **0.38** | **Synergy** |
| DP-C + Colistin | **0.38** | **Synergy** |
| DP-C + Meropenem | 0.67 | Additive/Indifference |
| DP-C + Ofloxacin | 0.75 | Additive/Indifference |
| DP-C + Rifampicin | 0.56 | Additive/Indifference |

**Notes.**
[a] FIC ≤ 0.5, synergistic; 0.5 < FIC ≤ 4.0, additive or unrelated; FIC > 4.0: antagonistic.

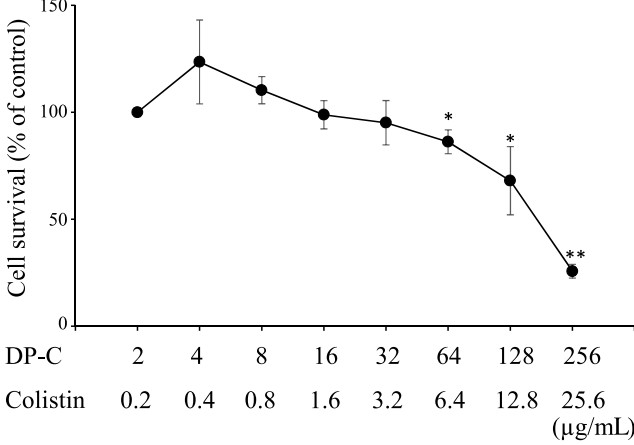

**Figure 1** **Cytotoxicity of colistin and antimicrobial peptides against HepG2 cells. HepG2 cells.** HepG2 cells were seeded at 3,000 cells/well in 96-well microtiter plates. After incubation for 48 h, DP-C and colistin were added and incubated for an additional 48 h. Cell viability was determined using a Cell Counting Kit-8, with colorimetric changes assessed at $OD_{450}$ with a microplate reader. Cell survival was expressed as a percentage of the control were obtained as mean ± SD of three independent experiments done in three replicates for each treatment. Significant differences between each concentration and the control (2 μg/mL DP-C and 0.2 μg/mL colistin) were statistically evaluated by Student's *t*-test (*$p < 0.05$, **$p < 0.01$).

concentrations showed synergistic effects of colistin and DP-C. The raw data of Fig. 1 are available in File S2.

## DISCUSSION

We added L-cysteine to the N- or C-terminus of [D(KLAKLAK)$_2$] in order to bind it with another protein like a single-chain antibody in the beginning of the experiment. This attachment yielded two compounds (DP-C and C-DP) with significantly greater antimicrobial activities against *A. baumannii, E. coli and P. aeruginosa* than the original DP. Similarly, the addition of L-cysteine to AMPs (Andersonin-Y1, HBcARD, buforinII or lysin) enhanced its antimicrobial activity, with higher membrane disruption activity than the

original peptide (*Chen et al., 2018*; *Pal et al., 2019*). The cysteine-derived cationic dipeptides lysine–cysteine, arginine–cysteine and histidine–cysteine presented antimicrobial activity, SEM analysis suggests that these dipeptides interact with cell walls to disrupt membrane integrity (*Tsai et al., 2020*). Whereas addition of an L-cysteine to the C-terminus of indolicidin, magainine or epinecidin-1 did not change their antimicrobial activity (*Chen et al., 2018*). It remains unclear what could be the mechanism by which the addition of cysteine to the N- or C-terminus to AMPs enhances antimicrobial activity. It is unlikely that this effect is simply due to peptide dimerization via cysteine disulfide bond formation because DP-C dimer showed the same MIC value as DP-C monomer against *P. aeruginosa* strains tested. The cysteine-rich region in Factor C receptors in the horseshoe crab specifically binds to bacterial lipopolysaccharides on Gram-negative bacteria (*Koshiba, Hashii & Kawabata, 2006*). The addition of L-cysteine to the N- or C-terminus of the peptide may have facilitated its binding to the bacterial membrane surface and form structures that disrupt their cell wall. The potential mechanism of efficacy enhancement by the attachment of cysteine residue to AMPs requires further investigation.

Systematic hybridization of two lead peptides from unrelated classes of AMPs showed no associations of net charge, charge density, and antipneumococcal activity among the hybrid peptides, although AMPs with higher hydrophobicity values have been reported to have greater antimicrobial activity against *Streptococcus pneumoniae* (*Le et al., 2015*). The peptides we tested showed a similar trend, when GRAVY was calculated, DPC showed higher hydrophobicity than DP. L-cysteine exhibited preferred antimicrobial activity against *S. aureus* compared with D-cysteine, whereas D-cysteine showed stronger antimicrobial activity against *E. coli*, *Listeria monocytogenes* and *Salmonella enteritis* (*Wang et al., 2019*). D-amino acid is highly resistant to proteolysis in bacteria, addition of D-cysteine instead of L-cysteine may be effective.

DP-C and C-DP had potent activity against MDR Gram-negative pathogens. The emergence and spread of these drug-resistant pathogens has become a serious worldwide public health problem (*Boucher et al., 2009*; *Tacconelli et al., 2018*). Carbapenem is the last resort $\beta$-lactam antibiotic administered to treat infections with drug-resistant Gram-negative pathogens. The development of new antibiotics against carbapenem-resistance pathogens is of top priority (*Tacconelli et al., 2018*). DP-C and C-DP also had antimicrobial activity against *E. coli* strains harboring the plasmid-mediated colistin resistance *mcr-1* gene. Colistin is a last line polycationic peptide antibiotic which is used to treat infections with carbapenem-resistant Gram-negative pathogens (*Paterson & Harris, 2016*). However, colistin-resistant *mcr-1* producers have emerged in humans and animals in China (*Liu et al., 2016*) and have spread worldwide. DP-C and C-DP, like DP, possess a positive charge and hydrophobic regions, suggesting that they target the lipid bilayer of the membrane and destroy it, causing loss of membrane potential and ultimately cell death (*McGrath et al., 2013*). These peptides had antimicrobial activity against Gram-negative but not Gram-positive pathogens. These peptides were also inactive against intrinsically colistin-resistant *S. marcescens*, indicating that the addition of positively charged 4-amino-4-deoxy-L-arabinopyranose 1 to lipopolysaccharide changes the membrane charge and prevents peptide binding. The combination of DP-C and colistin reduced effective doses of both

and may reduce peptide toxicity and colistin clinical nephrotoxicity. Although D-amino acid–based AMPs have been used clinically in the topical treatment of acne (*Gordon, Romanowski & McDermott, 2005*) but not yet for systemic infectious diseases. It should be rewarding to explore in the systemic treatment whether attachment of cysteine to the N- or C-terminus of AMPs could help broaden the spectrum and enhance the activity of AMPs against various drug-resistant microorganisms.

An algorithm predicting the effectiveness of in silico designed stapled AMPs, that are stable, active, and selective towards bacterial membranes in vivo, has enabled the modification of magainin II (Mag2) and other known AMPs (*Mourtada et al., 2019*). Modified Mag(i+4)1,15(A9K) was found to have MICs $< 4\,\mu g/mL$ for MDR *P. aeruginosa*, *A. baumannii*, and *E. coli* with concentrations as high as $\sim 100\,\mu g/mL$ having almost no red blood cell hemolytic activity. Combinations of colistin with DP-C and C-DP may achieve the same level of antimicrobial activity against these MDR bacteria, as well as widening the safety windows of both drugs. Furthermore, linking of chimeric DP-C and C-DP to macrocycles derived from polymyxin and colistin could have synergistic antimicrobial activity. Chimeric peptidomimetic antibiotics, in which a $\beta$-hairpin peptide macrocycle is linked to the macrocycles found in polymyxin and colistin, have shown potent antimicrobial activity against Gram-negative bacteria (*Luther et al., 2019*). These polymyxin and colistin-derived macrocycles targeting lipopolysaccharide were found to synergize with the macrocycles targeting $\beta$-barrel outer membrane proteins.

## CONCLUSIONS

Attachment of an L-cysteine residue to the N- or C-terminus of $[_D(KLAKLAK)_2]$ enhanced its antimicrobial activity against *P. aeruginosa, E. coli*, and *A. baumannii*. The combination of C-DP or DP-C and colistin had synergistic effects against MDR *P. aeruginosa*. In addition, DP-C and C-DP showed much stronger effects against MDR *A. baumannii* and *E. coli* than against *P. aeruginosa*.

## ACKNOWLEDGEMENTS

We thank Dr. Atsushi Ohnishi (Faculty of Pharmaceutical Sciences, Teikyo Heisei University) for providing us with HPLC analysis data of the DP-C dimer. We also thank Azumi Inaba for technical assistance and David Price for proofreading the manuscript.

### Funding

This investigation was supported by Grant-in-Aid for Scientific Research (C26460173), MEXT (Ministry of Education, Culture, Sports, Science and Technology)-Supported Program for the Strategic Research Foundation at Private Universities, 2014-2018 and a grant from the Naito Foundation. There was no additional external funding received for this study. The funders had no role in study design, data collection and analysis, decision to publish, or preparation of the manuscript.

## Grant Disclosures

The following grant information was disclosed by the authors:

Scientific Research: C26460173.

MEXT (Ministry of Education, Culture, Sports, Science and Technology)-Supported Program for the Strategic Research Foundation at Private Universities, 2014-2018 and a grant from the Naito Foundation.

## Competing Interests

The authors declare there are no competing interests.

## Author Contributions

- Maki K. Ohno and Isao Ishida conceived and designed the experiments, performed the experiments, analyzed the data, prepared figures and/or tables, authored or reviewed drafts of the paper, and approved the final draft.
- Teruo Kirikae analyzed the data, prepared figures and/or tables, authored or reviewed drafts of the paper, and approved the final draft.
- Eisaku Yoshihara analyzed the data, authored or reviewed drafts of the paper, and approved the final draft.
- Fumiko Kirikae performed the experiments, prepared figures and/or tables, and approved the final draft.

## Data Availability

The raw measurements are available in Supplemental Files.

## Supplemental Information

Supplemental information for this article can be found online at http://dx.doi.org/10.7717/peerj.10176#supplemental-information.

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
