# Peer review of "Addition of L-cysteine to the N- or C-terminus of the all-D-enantiomer [D(KLAKLAK)2] increases antimicrobial activities against multidrug-resistant Pseudomonas aeruginosa, Acinetobacter baumannii and Escherichia coli"

_PeerJ, doi:10.7717/peerj.10176_

## Round 0.1 · original submission · Minor Revisions

Two experts have reviewed the paper, and both found the study to be technically sound and had only minor comments on it. These relate mainly to statistical analyses, inclusion of HPLC profiles and mass spectra, and some small textual issues. Please address the points raised by the reviewers and submit a revised version of the manuscript.

Reviewer 1 ·

Basic reporting

The report is easy to read, relevant references are included.

The manuscript investigates how the addition of L-cysteine to a known antimicrobial peptide (in complete D-configuration) improves the antibacterial effect against certain Gram-negative pathogens and retains synergistic activity in combination with colistin. The scope of this work is rather limited by focusing on one peptide sequence. It would be interesting to explore whether incorporation of thiol residues into antimicrobial peptides is a more general strategy to improve the antibacterial effect of antimicrobial peptides.

Why was L-cysteine incorporated into the peptide sequence ? The D-cysteine likely would enhance metabolic stability. This is not clear.

Very little information was provided why the incorporation of L-cysteine enhances the antibacterial effect. ?

Experimental design

The experiments performed follow previously established standard protocols. Peptides were purchased and MIC, checkerboard studies and toxicity studies were performed using standard protocols.

Validity of the findings

I was unable to locate data that confirm complete disulfide formation of the dimeric peptides (MS spectra should be provided). It is usually standard to show the HPLC profiles of the purified peptides to demonstrate purity even when the peptides were purchased.

Additional comments

The scope is rather limited. It would be nice to see that this approach can be transferred to other antimicrobial peptides.

There is little understanding for the observed effect. In absence of mechanistic studies the results remain unexplained and do not enhance understanding.

Incorporation of D-amino acids into peptides can increase metabolic stability and proteolysis but why selecting L-cysteine instead of D-cysteine needs to be explained.

The synergistic effect of the thiol-modified peptides with other antibiotics could be investigated. For instance rifampicin and novobiocin are usually potentiated by antimicrobial peptides and it would be interesting to explore whether the effect is retained.

Reviewer 2 ·

Basic reporting

The manuscript is written in clear, unambiguous English with only minor typographical points observed as listed below. The abstract is clear and concise, and the introduction provides a good background to all areas that the manuscript investigates and is supported by the relevant literature. The manuscript is self-contained with all discussion points relevant to the results and the results are clearly relevant to the aims of the manuscript. Tables and figures are generally well presented but with some minor omissions that should be addressed (see elsewhere in the review). The manuscript is self-contained with all relevant information provided.

Minor typographical points:
Abstract: line 36 “C-DP or DP-C” should be changed to “C-DP and DP-C”

Abstract: lines 39-40 “overexpressing the efflux” should be changed to “overexpressing efflux”

Abstract: line 49 “showed much stronger effects against” this should be altered to be more specific in describing what the stronger effect was e.g. stronger antimicrobial activity

Introduction: line 66 “thought to bind to cytoplasmic membrane” consider “thought to bind to the cytoplasmic membrane”

Results line 171: “but DP not against OCR1” suggested change to “but showed no antimicrobial activity against OCR1 within the tested concentration range” or similar to improve clarity.

Experimental design

This manuscript provides original primary research within the aims and scope of PeerJ. The human cell line HepG2 is described correctly with ATCC number provided. Methods are described with sufficient detail and information to replicate experiments. The research question well defined, relevant & meaningful. It is stated how research fills an identified knowledge gap to further improve the efficacy of antimicrobial peptides.

Statistical analyses should be performed when comparing the difference between MICs of the different AMPs to show that the increase in sensitivity is statistically significant.

Statistical analyses should also be used to determine at what concentration of DP-C and colistin cell line viability is significantly decreased.

Minor points:
The authors state that peptides were synthesised to >95% purity. Can the authors state how the >95% purity of peptides was measured or if synthesised externally explicitly state that the synthesis was conducted by the named external company.

Lines 111-113 and 115-117: description of what the strains do and are for should not be in the methods section, they should be in the results section.

Table 1: It should be stated clearly within the table that the antibiotic concentrations shown are the MIC concentrations.

Validity of the findings

Raw data has been provided for all human cell line assays in the form of percentage viability, an appropriate number of replicates is provided with mean, SD and SEM are calculated. This information is however unclear in figure 1, number of replicates, and statement of whether SD or SEM is used to create the graph are needed in the legend.

The conclusions reflect the results presented with little speculation.

---

## Round 0.2 · Minor Revisions

The reviewer would still like to see amendments to the legend of Figure 1. Please make these and resubmit a revised version.

Reviewer 2 ·

Basic reporting

Requested improvements were made to the text in the revised mansucript.

Experimental design

No comment

Validity of the findings

No comment

Additional comments

I cannot see the changes that the authors said they made to the legend of Figure 1 in their rebuttal letter - Raw data has been provided for all human cell line assays in the form of percentage viability, an appropriate number of replicates is provided with mean, SD and SEM are calculated. This information is however unclear in figure 1, number of replicates, and statement of whether SD or SEM is used to create the graph are needed in the legend.

This information is now found in the methods section, however can the authors include it in the figure legend for completeness and ease of interpreting the data (e.g. N=3 +/- SD). Also the meaning of the * in the figure should be clearly described in the figure legend (e.g. Students t-test *p<0.05).

---

## Round 0.3 · accepted · Accept

The concerns of the reviewers have been addressed and manuscript is now acceptable for publication.